# LacTok: Latent Consistency Tokenizer for High-resolution Image Reconstruction and Generation by 256 Tokens

## Abstract

Image tokenization has significantly advanced visual generation and multimodal modeling, particularly when paired with autoregressive models. However, current methods face challenges in balancing efficiency and fidelity: high-resolution image reconstruction either requires an excessive number of tokens or compromises critical details through token reduction. To resolve this, we propose Latent Consistency Tokenizer (LacTok) that bridges discrete visual tokens with the compact latent space of pretrained Latent Diffusion Models (LDMs), enabling efficient representation of 1024×1024 images using only 256 tokens—a 16× compression over VQGAN. LacTok integrates a transformer encoder, a quantized codebook, and a latent consistency decoder. Direct application of LDM as the decoder results in color and brightness discrepancies; thus, we convert it to latent consistency decoder, reducing multi-step sampling to 1-2 steps for direct pixel-level supervision. Experiments demonstrate LacTok's superiority in high-fidelity reconstruction, with 10.8 reconstruction Fréchet Inception Distance on MSCOCO-2017 5K benchmark for 1024×1024 image reconstruction. We also extend LacTok to a text-to-image generation model, LacTokGen, working in autoregression. It achieves 0.73 score on GenEval benchmark, surpassing current state-of-the-art methods.

## 1 Introduction

Image tokenizer Van Den Oord et al. (2017); Yu et al. (2024b) aims to convert images from their raw pixel-based representations into discrete visual tokens, which can then be used to reconstruct the original image through its corresponding decoder. This approach has garnered significant attention due to its crucial role in image generation, particularly in autoregressive models Sun et al. (2024); Tian et al. (2024); Wang et al. (2024) and masked transformers Chang et al. (2023; 2022); Ding et al. (2022); Xie et al. (2024a).

A representative approach, VQGAN Yu et al. (2022) learns a codebook to quantize continuous embeddings of a 256×256 image into 256 discrete tokens using a spatial downsampling ratio of 16×—a standard configuration in recent works Mizrahi et al.; Qu et al. (2024); Shi et al. (2022); Sun et al. (2024). When it comes to high-resolution image reconstruction or generation, e.g., 1024×1024 pixels, requires predicting 4096 tokens, creating substantial challenges in both computational efficiency and model optimization. This lengthy token sequence complicates downstream applications, such as integrating visual tokens into large multimodal models for interleaved text and image understanding and generation. While recent advancements explore token compression through residual codebooks Lee et al. (2022b); Tian et al. (2024) and 1D tokenization Kim et al. (2025); Yu et al. (2024b); Li et al. (2025); Bachmann et al. (2025) for 256×256 images, these approaches have not substantially addressed the challenges of higher-resolution image reconstruction and generation. Recent latent diffusion models (LDMs) Esser et al. (2024b); Labs (2024); Podell et al. (2024) have demonstrated remarkable success in 1024×1024 image generation by operating in low-dimensional latent spaces. This raises a compelling question: Can discrete tokens be aligned with the compact latent space of LDMs to leverage their powerful decoders for high-fidelity reconstruction and generation use?

In this paper, we introduce a Latent Consistency Tokenizer (LacTok), comprising a transformer encoder, a quantized vector codebook, and a latent consistency decoder. Our key insight is to align

discrete visual tokens with the compact latent space of pretrained LDMs, enabling efficient representation of 1024×1024 images with only 256 tokens—a 16× reduction compared to VQGAN. Inspired by ControlNet Zhang et al. (2023), we first employ a latent diffusion decoder through copying adaptive blocks in LDM with zero convolution connecting copy and raw LDM. During training, LacTok is optimized using diffusion objectives with progressive resolution scaling from 512×512 to 1024×1024. However, relying solely on diffusion loss results in reconstructed images with noticeable discrepancies in color and brightness. To address this, we introduce latent consistency models Ren et al. (2024); Xie et al. (2024b) to the decoder, converting the multi-step sampling process into one or two steps. It enables more direct supervision and improving reconstruction fidelity. Moreover, we extend LacTok to text-to-image (T2I) generation model (LacTokGen) by training an autoregressive transformer, which efficiently generates these compact token sequences through text-instructed autoregressive prediction.

We conducted comprehensive experiments to evaluate LacTok on several validataset datasets. With 256 tokens to reconstruct 1024×1024 pixel images, LacTok achieves 10.80 reconstrution Fréchet Inception Distance (rFID) score Heusel et al. (2017) on MSCOCO-2017 5K validation dataset Lin et al. (2014), significantly outperforming SeedTok Ge et al. (2023), TiTok Yu et al. (2024b), LlmaGen Sun et al. (2024), and FlexTok Bachmann et al. (2025). On ImageNet benchmark Deng et al. (2009), LacTok gets 2.78 rFID, surpassing SeedTok and LlamaGen, comparable to TiTok and FlexTok, but with much better Peak Signal-to-Noise Ratio, Structural Similarity Index Measure, and Learned Perceptual Image Patch Similarity than TiTok and FlexTok. On MJHQ-5K dataset Playgroundai (2023) and FLUX-5k dataset synthesized by FLUX.1-dev Labs (2024), LaTok obtains noticeably lower rFID than other tokenizers. For text-to-image task, LacTokGen obtains 0.73 score on GenEval benchmark Ghosh et al. (2023), superior to other diffusion and autoregressive models, e.g., 0.62 by SD3 Esser et al. (2024a), 0.56 by HART Tang et al. (2024), 0.53 by Show-o Xie et al. (2024a), and 0.32 by LlamaGen Sun et al. (2024).

Our contributions are threefold:

- We propose LacTok, an image tokenizer that bridges discrete visual tokens with the latent space of pretrained LDMs, demonstrating the ability to reconstruct and generate 1024×1024 images with only 256 tokens.

- To alleviate color and brightness discrepancies caused by diffusion objective in reconstruction, we integrate latent consistency model into latent diffusion decoder with direct pixel-level supervision through efficient few-step sampling.

- Built on LacTok, LacTokGen can generate high-quality 1024×1024 images via text-guided autoregressive token prediction. It achieves leading performances on multiple synthesis benchmarks.

## 2 RELATED WORK

### 2.1 IMAGE TOKENIZATION

Variational Autoencoders (VAEs) Kingma (2013) represent a significant advancement in the field by learning to map inputs to a distribution. Building upon this foundation, VQVAEs Van Den Oord et al. (2017) learn discrete representations that form a categorical distribution. This process is further improved in VQGAN Esser et al. (2021), which enhances the training process through the integration of adversarial training techniques. The transformer architecture within autoencoders is explored in ViT-VQGAN Yu et al. (2022). RQ-VAE Lee et al. (2022b), introduces residual quantization to the VAE framework, recursively quantizes the feature map in a coarse-to-fine manner, allowing for a precise approximation of the feature map with a fixed codebook size. In a different vein, MAGVIT-v2 Yu et al. (2024a), FSQ Mentzer et al. (2024), BSQ-ViT Zhao et al. (2024) propose lookup-free quantization, presenting an alternative approach that bypasses traditional lookup mechanisms. Visual autoregressive modeling Tian et al. (2024) exploits next-scale prediction for image reconstruction, while TiTok Yu et al. (2024b) and MergeVQ Li et al. (2025) compress token number by 1D tokens to represent the same image. DiVAE Shi et al. (2022), SEED tokenizer (SeedTok) Ge et al. (2023) and FlexTok Bachmann et al. (2025) incorporate diffusion models into tokenizer for image tokenization. Nevertheless, they either do not validate their efficacy on large-scale text-to-image task, or generate undesirable images. VILA-U Wu et al. (2024), TokenFlow Qu et al. (2024),

and TA-TiTok Kim et al. (2025) introduce text supervision to enhance semantic information for discrete tokens. However, they can not reconstruct or generate high-resolution image details by small number of tokens. Motivated by LDMs Esser et al. (2024a); Labs (2024); Podell et al. (2024) which are able to generate high-resolution, high-quality images, we explore effective method to leverage the pretrained LDM as decoder to reconstruct high-resolution images.

## 2.2 Tokenized Image Generation

Image tokenization has become a powerful technique for image generation, allowing images to be represented as discrete tokens that can be manipulated and generated using various modeling approaches. Two prominent methodologies in this domain are the masked-transformer style and the autoregressive style. In the masked-transformer style, MaskGIT Chang et al. (2022) utilizes a bidirectional transformer decoder. During training, the model predicts randomly masked tokens and iteratively refines the image at inference. Other notable works in this category include Chang et al. (2023); Lee et al. (2022a); Lezama et al. (2022a;b). Conversely, the autoregressive style involves predicting all tokens of an image prediction Esser et al. (2021); Lu et al. (2023); Wang et al. (2024); Han et al. (2025). HART Tang et al. (2024) uses hybrid tokenizers for image generation. Other works Jin et al. (2024); Team (2024); Wu et al. (2024); Zhan et al. (2024) integrate discrete tokenizers with large language models to generate images. LlamaGen Sun et al. (2024) stands out as a simple yet effective generation method that adopts the autoregressive approach, but it fails to generate satisfactory images. We extend LacTok to T2I generation model by training an autoregressive model to predict the next discrete tokens produced by the tokenizer. Thanks to the excellent performance of LacTok to reconstruct images, our model is able to generate superior images.

# 3 Preliminary

## 3.1 VQGAN.

VQGAN Esser et al. (2021) typically consists of encoder $Enc$, quantizer $Q$, and decoder $Dec$. Given an image $\mathbf{x} \in \mathbb{R}^{H \times W \times 3}$, where $H, W$ denotes image height and width, respectively, $Enc$ firsts extracts its latent embeddings $\mathbf{G} = Enc(\mathbf{x}) \in \mathbb{R}^{H/f \times W/f \times D}$, effectively reducing the spatial dimensions by a factor of $f$. Then, $Q$ maps each embedding $\mathbf{g} \in \mathbb{R}^D$ in $\mathbf{G}$ with $D$ representing embedding dimension to the nearest code $c_i$ in a learnable codebook $\mathbb{C} \in \mathbb{R}^{N \times D}$, where $N$ is the codebook size. Mathematically, this can be formulated as:

$$Q(\mathbf{g}) = \mathbf{c}_{Tok}, \quad Tok = \arg \min_{j \in \{1,2,...,N\}} \|\mathbf{g} - \mathbf{c}_j\|_2^2. \tag{1}$$

The mapped feature vectors $\mathbf{C} \in \mathbb{R}^{H/f \times W/f \times D}$ are calculated by $Q(\mathbf{G})$, while decoder converts $\mathbf{C}$ to image $\hat{\mathbf{x}}$ through $Dec(\mathbf{C})$).

## 3.2 Diffusion Models.

Diffusion model Ho et al. (2020) is composed of a forward diffusion process and a reverse denoising process. Forward process gradually adds random noise to clean data $\mathbf{x}_0$ and diffuses it into pure Gaussian noise as:

$$\mathbf{x}_t = \alpha_t \mathbf{x}_0 + \beta_t \epsilon, \quad t \in [0, T], \tag{2}$$

where $\epsilon \in \mathcal{N}(0, \mathbf{I})$, $\alpha_t$, $\beta_t$ are scheduler coefficients with $\alpha_t^2 + \beta_t^2 = 1$, and $T$ is the ending time. During the training stage, DMs usually minimizes diffusion loss $\mathcal{L}_{DF}$ via neural network $\epsilon_\theta$ predicting noise as:

$$\mathcal{L}_{DF} = \|\epsilon_\theta(\mathbf{x}_t, t) - \epsilon\|_2^2. \tag{3}$$

Denoising process reverses the diffusion process to create clean samples from the Gaussian noise.

To reduce resource consumption, Latent Diffusion Models Esser et al. (2024b); Labs (2024); Podell et al. (2024) transform pixel image $\mathbf{x}_0$ into latent embedding $\mathbf{z}_0$ via VAE, and perform diffusion and denoising operations in latent space. Sequentially, VAE decoder converts predicted $\mathbf{z}_0$ back to the pixel image. LDMs has shown great success in text-to-image, image-to-image tasks, but they have not been effectively explored for building discrete tokenizer. In this study, we investigate effective method which enables LDMs to construct discrete tokenizer for high-resolution image reconstruction and generation.

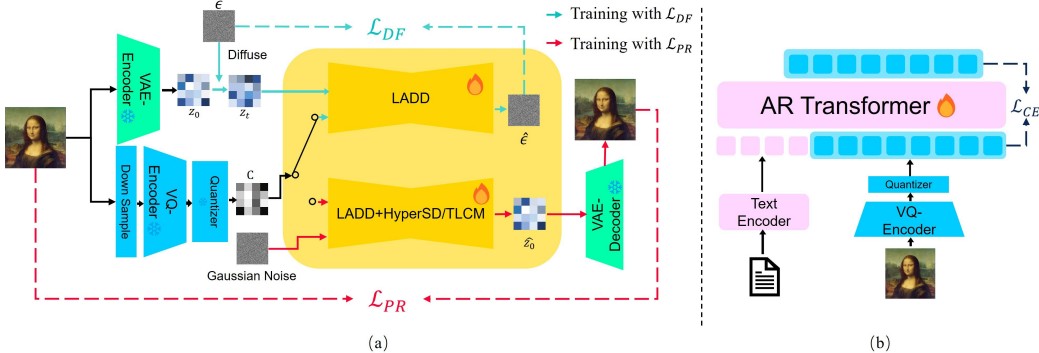

(a)                                                                                          (b)

Figure 1: (a) Overview of LacTok. The input image is sequentially processed by downsampler, encoder and quantizer into condition features. It also goes through the VAE Encoder in pretrianed LDM to produce latent $z_0$, which will be diffused to produce $z_t$. LADD takes $C$, $z_t$ and $t$ as input. In the first phase, we apply $\mathcal{L}_{DF}$ to train LADD. In the second phase, we introduce acceleration models to LADD and replace $z_t$ with Gaussian noise to perform one or two step inference, which allows us to train LADD with pixel reconstruction loss $\mathcal{L}_{PR}$. For simplicity, we omit the time step $t$. (b) Illustration of autoregressive text-to-image generation with LacTokGen.

## 4 METHODOLOGY

Conventional discrete tokenizers Esser et al. (2021); Sun et al. (2024); Van Den Oord et al. (2017); Yu et al. (2022); Li et al. (2025) encode images into discrete tokens and decode the tokens into original images in pixel space. These works transform text-to-image generation into next-prediction task, which predict image tokens by autoregressive model conditioned on the text input. However, these tokenizers need a large number of tokens to generate high-resolution images, leading to prohibitive training and inference costs. Moreover, these tokenizers can not faithfully reconstruct images with intricate, high-frequency details such as human faces. Such drawback sets a low ceiling for text-to-image generation task.

We are targeting at constructing an innovative tokenizer for high-resolution image reconstruction and generation via a small number of discrete tokens. As information loss inevitably occurs after encoding and quantization, which is a form of lossy compression. The decoder's task is to predict an image as close as possible to the input from limited information. To this end, we propose Latent Consistency Tokenizer (LacTok) which is capable of reconstructing and producing high-quality images with only 256 tokens. Unlike the existing tokenizers that operate image encoding and decoding in pixel space, LacTok models decoding procedure in latent space through unleashing the potential of latent diffusion model, which further makes use of latent consistency models Ren et al. (2024); Xie et al. (2024b), to enforce pixel reconstruction. Upon LacTok, we build an autoregressive model for high-resolution image generation driven by text conditions. Figure 1 illustrates the overview of the proposed method.

### 4.1 LATENT DIFFUSION RECONSTRUCTION

Considering that latent diffusion models Esser et al. (2024a); Podell et al. (2024) are powerful to synthesize high-quality images, we propose latent diffusion decoder (LADD), denoted as $f_\theta$, to build discrete tokenizer. The core thought of LADD is to predict the latent representation $\mathbf{z}_0$ conditioned on the pixel image $\mathbf{x}_0$. To fulfill this idea, the raw image $\mathbf{x}_0 \in \mathbb{R}^{H \times W \times 3}$ is converted into latent code $\mathbf{z}_0 \in \mathbb{R}^{H/8 \times W/8 \times C}$ using VAE encoder of pretrained LDM, where $H = W \in \{512, 1024\}$. To predict $\mathbf{z}_0$, diffusion loss is used to train LADD as:

$$\mathcal{L}_{DF} = \|f_\theta(\mathbf{z}_t, \mathbf{C}, t) - \epsilon\|_2^2, \tag{4}$$

where $\mathbf{z}_t$ is obtained by forward diffusion procedure using Eq. (2) in latent space. The condition $\mathbf{C}$ is obtained by sequentially encoding $\mathbf{x}_0$ using $Enc$ and the quantizer $Q$ through the Eq. (1). To reduce token number, $\mathbf{x}_0$ is first downsampled into the resolution of $H' \times W'$, where $H' = W' \in \{224, 256, 288\}$. To promote training's stability, we draw inspiration from ControlNet Zhang et al. (2023) to design LADD structure. Specifically, LADD freezes the parameters of pretrained

LDM and simultaneously clones some blocks of LDM to a trainable copy. Zero convolution ($ZC$) is utilized to connect trainable copy and raw LDM. The output $\mathbf{O}$ of LADD block at timestep $t$ is computed as:

$$\mathbf{O} = ZC(F_{train}(z_t, \mathbf{C}, t)) + F(z_t, t), \tag{5}$$

where $F$ and $F_{train}$ denote the frozen LDM block and trainable block in trainable copy, respectively.

To train our tokenizer and save training resource, the visual encoder and quantilizer are initialized using pretrained LlamaGen tokenizer. Next, we freeze the visual encoder and quantilizer, enabling only LADD to undergo training.

## 4.2 PIXEL RECONSTRUCTION

The diffusion loss enables the decoder to reconstruct the original image through multi-step sampling. However, we have empirically observed that the reconstructed images exhibit discrepancies in color and brightness compared to the original images. To address this issue, we introduce a pixel reconstruction loss $\mathcal{L}_{PR}$ that compels the decoded image to recover the original image. Since diffusion model requires multi-step sampling to generate images, it is unable to directly minimize $\mathcal{L}_{PR}$, which consumes a significant amount of GPU memory and may lead to gradient explosion. Consistency model (CM) can enable LDMs to generate images with few-step inference. Therefore, we leverage CM to assist LADD in achieving rapid image reconstruction. As HyperSD Ren et al. (2024) and Training-efficient Latent Consistency Model (TLCM) Xie et al. (2024b) show state-of-the-art performance, both of which are CMs for LDM'acceleration, they are integrated into LADD to reconstruct image with few steps. Sequentially, $\mathcal{L}_{PR}$ can be used to optimize our tokenizer as gradient is easily propagated into decoder. One-step sampling is used to reconstruct clean latent code $\hat{\mathbf{z}}_0$ when HyperSD is merged into LADD, because it can generate high-quality image with one step. Two-step sampling is leveraged to restore $\hat{\mathbf{z}}_0$ when incorporating TLCM into LADD since it needs at least two steps, where stop-gradient operation is adopted for the first iteration. Both models accept pure Gaussian noise as initial latent code. The reconstruction loss $\mathcal{L}_{PR}$ is:

$$\mathcal{L}_{PR} = \mathcal{L}_P(Dec(\hat{\mathbf{z}}_0), \mathbf{x}_0) \tag{6}$$

where $Dec(.)$ represents the pretrained VAE decoder in LDM, $\mathcal{L}_P$ is a perceptual loss from LPIPS Zhang et al. (2018). We use LacTok-H and LacTok-T to represent that HyperSD and TLCM are utilized in LADD to minimize $\mathcal{L}_{PR}$, respectively.

## 4.3 TEXT-TO-IMAGE GENERATION

In order to unleash the value of tokenizer, we leverage LacTok to build text-to-image generation model (LacTokGen). LacTokGen is implemented through an autoregressive models $P_\theta$ with $\theta$ denoting parameters. A text encoder is used to extract text feature $\mathbf{f}_{text}$, which is projected by an additional MLP to match the dimension of autoregressive models. Cross entropy loss $\mathcal{L}_{CE}$ is applied to train autoregressive model as:

$$\mathcal{L}_{CE} = -\sum_{i=1}^{L} \log P_\theta(Tok_{i+1}|Tok_{i:1}, \mathbf{f}_{text}), \tag{7}$$

where $L$ is token number to represent a image. As classifier-free guidance (CFG) Ho & Salimans (2022) is critical to generate high-quality image in LDM, we adopt it in our models. During training, the conditional is randomly replaced by a null unconditional embedding. During inference stage, the logit $\ell_g$ is computed as $\ell_g = \ell_u + s(\ell_c - \ell_u)$ for every token, where $\ell_c$ represents the conditional logit, $\ell_u$ denotes the unconditional logit, and $s$ is the scaling factor for CFG.

## 4.4 DATA CONSTRUCTION

It is well known that high-quality data is necessary to train LDM, but it is hard to access a mass of real data. To deal with this challenge, FLUX.1-dev Labs (2024) is used to produce data as it can generate superior images. The text input for FLUX.1-dev is from LAION-400M Schuhmann et al. (2021). Totally, we synthesize 30M images, and synthetic images and LAION-Aesthetics-6.5+ constitute the data source. Since raw caption is too simple to describe image content, we use Qwen2.5-VL-72B Bai et al. (2025) to generate new caption according to image and raw caption. The new caption

Table 1: The reconstruction performance of LacTok on ImageNet, MSCOCO-2017 5K validation dataset. All the tokenizers are trained on ImageNet, except for those specifically marked with * or +. * denotes the tokenizer is trained on our constructed data. + denotes the tokenizer is trained on on CC3M Sharma et al. (2025), Unsplash Chesser & Carbone (2023), LAION-COCO Christoph et al. (2022), and COCO Chen et al. (2015)

| Methods | ImageNet | | | | MSCOCO-2017 | | | |
|---|---|---|---|---|---|---|---|---|
| | rFID↓ | P↑ | S↑ | L↓ | rFID↓ | P↑ | S↑ | L↓ |
| SeedTok+ Ge et al. (2023) | 15.65 | 9.39 | nan | 0.69 | 23.28 | 9.37 | nan | 0.72 |
| TiTok-S-128 Yu et al. (2024b) | 2.32 | 16.97 | 0.51 | 0.49 | 12.31 | 16.47 | 0.50 | 0.51 |
| LlamaGen Sun et al. (2024) | 3.17 | **19.94** | 0.64 | 0.41 | 11.23 | **19.53** | **0.64** | 0.43 |
| FlexTok Bachmann et al. (2025) | **2.00** | 18.02 | 0.58 | 0.46 | 13.08 | 17.23 | 0.56 | 0.49 |
| LacTok-H(Ours) | 2.78 | 19.80 | **0.65** | **0.39** | **10.80** | 19.28 | **0.64** | **0.41** |
| LacTok-H*(Ours) | 9.04 | 19.02 | **0.65** | 0.43 | 16.93 | 18.20 | **0.64** | 0.43 |
| LacTok-T*(Ours) | 8.71 | 18.80 | **0.65** | 0.43 | 15.05 | 17.98 | **0.64** | 0.42 |

Table 2: The reconstruction performance of various methods on MJHQ-5K and FLUX-5K datasets.

| Methods | MJHQ-5K | | | | FLUX-5K | | | |
|---|---|---|---|---|---|---|---|---|
| | rFID↓ | P↑ | S↑ | L↓ | rFID↓ | P↑ | S↑ | L↓ |
| SeedTok+ Ge et al. (2023) | 23.87 | 8.39 | nan | 0.73 | 25.85 | 8.88 | nan | 0.74 |
| TiTok-S-128 Yu et al. (2024b) | 14.17 | 16.42 | 0.54 | 0.50 | 15.09 | 16.29 | 0.59 | 0.47 |
| LLamaGen Sun et al. (2024) | 13.26 | **19.24** | 0.68 | 0.41 | 13.28 | **19.05** | 0.72 | 0.38 |
| FlexTok Bachmann et al. (2025) | 16.17 | 17.21 | 0.60 | 0.47 | 16.14 | 17.68 | 0.67 | 0.43 |
| LacTok-H(Ours) | **11.34** | 19.16 | 0.68 | **0.38** | 12.45 | 18.68 | 0.73 | 0.37 |
| LacTok-H*(Ours) | 14.02 | 18.04 | 0.68 | 0.39 | 12.18 | 18.70 | **0.74** | 0.35 |
| LacTok-T*(Ours) | 13.32 | 17.81 | **0.69** | **0.38** | **11.04** | 18.90 | **0.74** | **0.34** |

has to describe main objects and their attributes, as well as spatial relationship among the objects, within 20 words. As some images do not meet human preferences, ImageReward (IR) Xu et al. (2023)$> 0.9$ and Multi-dimensional Preference Score (MPS) Zhang et al. (2024)$> 12.0$ are used to select desirable image-caption pairs from data source. Finally, we build 20M high-quality text-image pairs. The examples of the image-caption pairs are shown in Figure 4.

## 5 EXPERIMENTAL RESULTS

Please refer to Section A for implementation details.

### 5.1 IMAGE RECONSTRUCTION

We employ Peak Signal-to-Noise Ratio (P), Structural Similarity Index Measure (S) to evaluate the image similarity between reconstructed and raw images. Concurrently, we evaluate the distribution discrepancy by reconstruction Fréchet Inception Distance score (rFID) Heusel et al. (2017). Learned Perceptual Image Patch Similarity (L) Zhang et al. (2018) is further used to measure perceptual similarity as it performs excellently in terms of human visual perception. The validation is conducted on ImageNet 50K validation set Deng et al. (2009), MSCOCO-2017 5K validation set Lin et al. (2014), MJHQ-5K validation set which is randomly sampled from MJHQ-30K dataset Playgroundai (2023) and FLUX-5K validation dataset which is synthesized by FLUX.1-dev. All the images are resized to $1024 \times 1024$ size for evaluation as we aim at high-resolution image reconstruction.

Tables 1 and 2 list the reconstruction performance of the different tokenizers, where all tokenizers use 256 tokens except that SeedTok and TiTok-S-128 adopt 32 tokens and 128 tokens, respectively. It can be observed that our LacTok-H demonstrates strong performance compared to other tokenizers. SeedTok Ge et al. (2023) presents poor performance across all the datasets, indicating the difficulty of image reconstruction using a pretrained LDM. Our approach overcomes this by integrating diffusion and pixel reconstruction losses, leading to remarkable improvements. While LacTok-H shows a slightly higher rFID than FlexTok Bachmann et al. (2025) and TiTok Yu et al.

(2024b) on object-centric ImageNet, it achieves superior scores on P, S, and L metrics. More importantly, LacTok-H exhibits substantially better performance on datasets with complex scenes, such as MSCOCO-2017, MJHQ-5K, and FLUX-5K. Furthermore, LacTok-H surpasses LlamaGen Sun et al. (2024) on most metrics (rFID, S, L) across four datasets, with only a comparable P value. This advantage stems from our use of a pretrained LDM enhanced by HyperSD, which excels at reconstructing fine-grained details like human faces.

Moreover, the rFID of LacTok-H* and LacTok-T* trained on our constructed data is higher than LacTok-H, because data distribution of the constructed images dramatically deviate from those of ImageNet, MSCOCO, and MJHQ sets. When evaluated on FLUX-5K dataset, LacTok-H* and LacTok-T* get better rFID score than other methods. The performance of LacTok-H* concerning rFID, L are improved by LacTok-T*, thanks to its ability to reconstruct intricate, high-frequency by increasing the sampling steps.

Figure 2 illustrates the visual comparisons of image reconstruction for different tokenizers. We can observe that the proposed LacTok-H significantly improves reconstruction performance than SeedTok, TiTok, FlexTok, and LlamaGen, especially in image details such as faces. Besides, it can be seen that the images reconstructed by LacTok-H* and LacTok-T* show better quality than other methods. The result also indicates that lower rFID on ImageNet does not necessarily indicate better reconstruction with respect to human preference.

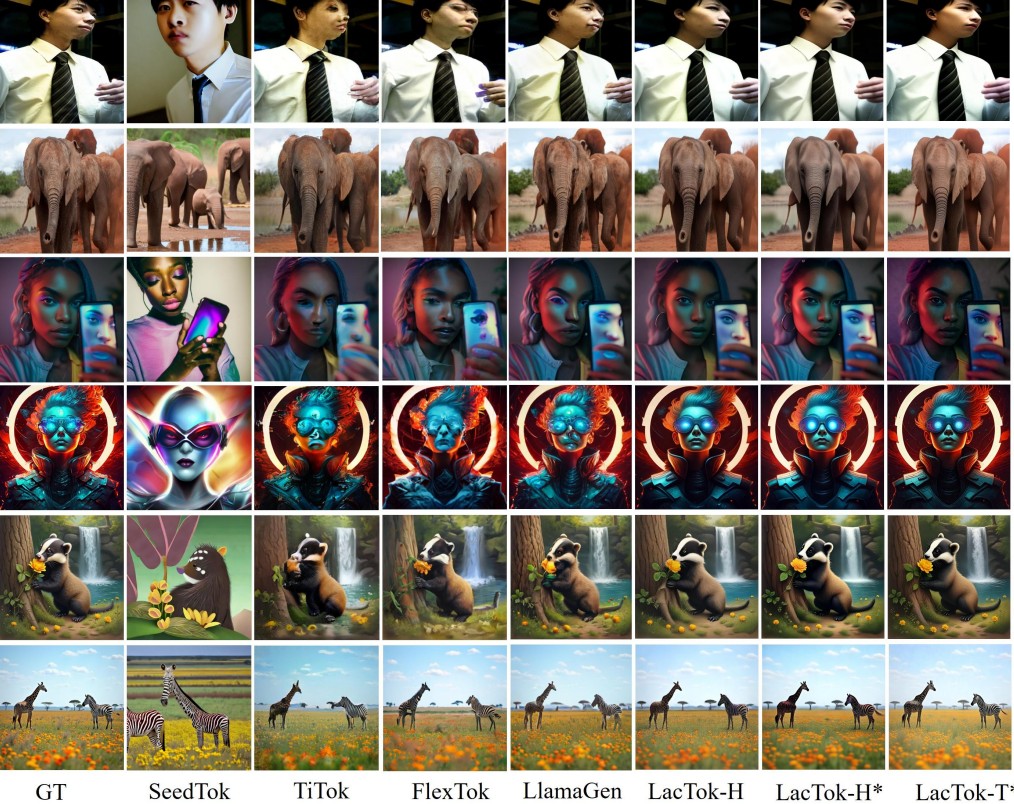

| GT | SeedTok | TiTok | FlexTok | LlamaGen | LacTok-H | LacTok-H* | LacTok-T* |

Figure 2: Visual comparisons of images reconstruction for different methods. LacTok can reconstruct higher-quality images than SeedTok, TiTok, FlexTok, and LlamaGen, especially in detail reconstruction.

## 5.2 TEXT-TO-IMAGE GENERATION

To evaluate the conditional generation performance of LacTok, we perform large-scale text-to-image experiments using LacTokGen. The evaluation is conducted on GenEval benchmark Ghosh et al. (2023) and MSCOCO-2017 5K validation dataset Lin et al. (2014). GenEval benchmark is used

Table 3: The performance of our text-to-image models on GenEval and MSCOCO-2017 5K validation dataset.

| Methods | GenEval↑ | | | | MSCOCO-2017↑ | | |
|---|---|---|---|---|---|---|---|
| | Two Obj | Position | Color Attri | **Overall** | IR | MPS | HPSv2 |
| Diffusion Models | | | | | | | |
| SD1.5 Rombach et al. (2022b) | 0.38 | 0.04 | 0.06 | 0.43 | 0.16 | 10.08 | 0.285 |
| SDXL Podell et al. (2024) | 0.74 | 0.15 | 0.23 | 0.55 | 0.82 | 11.9 | 0.295 |
| SD3 Esser et al. (2024b) | 0.74 | 0.34 | 0.36 | 0.62 | **1.00** | 12.59 | 0.303 |
| FLUX.1-dev Labs (2024) | **0.85** | 0.21 | 0.45 | 0.68 | **1.00** | **12.97** | 0.306 |
| AutoRegressive Models | | | | | | | |
| LlamaGen Sun et al. (2024) | 0.34 | 0.07 | 0.04 | 0.32 | 0.29 | 9.56 | 0.273 |
| HART Tang et al. (2024) | - | - | - | 0.56 | 0.66 | 11.69 | 0.298 |
| Show-o Xie et al. (2024a) | 0.52 | 0.11 | 0.28 | 0.53 | 0.95 | 10.58 | 0.277 |
| LacTokGen-H | 0.67 | 0.49 | 0.50 | 0.71 | 0.86 | 12.22 | 0.298 |
| LacTokGen-H* | 0.69 | 0.51 | 0.51 | 0.72 | 0.88 | 12.30 | 0.302 |
| LacTokGen-T* | 0.72 | **0.53** | **0.51** | **0.73** | 0.90 | 12.38 | **0.304** |

Table 4: Ablation study of key components in LacTok. VQ-LADD means substituting decoder in LlamaGen with LADD, and using 25-step DDIM Song et al. (2020) to reconstruct images.

| Methods | rFID↓ | P↑ | S↑ | L↓ |
|---|---|---|---|---|
| LlamaGen | 13.26 | **19.24** | **0.68** | 0.41 |
| VQ-LADD | 12.72 | 16.53 | 0.63 | 0.47 |
| VQ-LADD+HyperSD | 14.71 | 16.64 | 0.64 | 0.47 |
| VQ-LADD+TLCM | 14.40 | 16.82 | 0.65 | 0.45 |
| LacTok-H | **11.34** | 19.16 | **0.68** | **0.38** |

to evaluate compositional image properties, such as spatial relations and attribute binding. On MSCOCO dataset, IR, MPS, and HPSv2 Wu et al. (2023) are used to assess human preference of the generated image.

Table 3 summaries the performance of several models on two validation datasets. LacTokGen-T* outperforms LlamaGen on GenEval benchmark by 0.41 points and achieves substantially better results across all metrics on MSCOCO-2017. This result demonstrates the superiority of our model in terms of image quality, which derives from LacTokGen-T*'s capacity to effectively leverage the pretrained Latent Diffusion Model (LDM) for rendering fine-grained details. LacTokGen-T* also surpasses SDXL by 0.18 on GenEval and improves all metrics on MSCOCO-2017. This gain stems from two factors: (1) our tokenizer efficiently represents high-resolution images with discrete tokens, simplifying autoregressive modeling compared to diffusion-based training; and (2) the use of high-quality training data for the autoregressive model. Moreover, LacTokGen-T* significantly outperforms other autoregressive models like Show-o Xie et al. (2024a) and HART Tang et al. (2024) on both benchmarks, as these rely on simple CNN or transformer decoders that struggle to reconstruct fine details and spatial relationships. Notably, LacTokGen-T* also beats SD3 Esser et al. (2024b) and FLUX.1-dev on GenEval, due to its use of informative captions that well describe compositional scenes. Additionally, LacTokGen-H* improves upon LacTokGen-H through better training data, suggesting that ImageNet reconstruction metrics do not fully reflect generative capability. The performance of LacTokGen-H* is further improved by LacTokGen-T*, which again verifies that TLCM is more powerful than HyperSD to help decoder generate high-quality images.

Figure 5 compares the visual results of different models. we can observe that the images generated by LacTokGen-T* enjoy better text-image alignment and higher human preference than SD1.5 Rombach et al. (2022a), SDXL, HART Tang et al. (2024), Show-o Xie et al. (2024a), and LlamaGen, especially in facial generation.

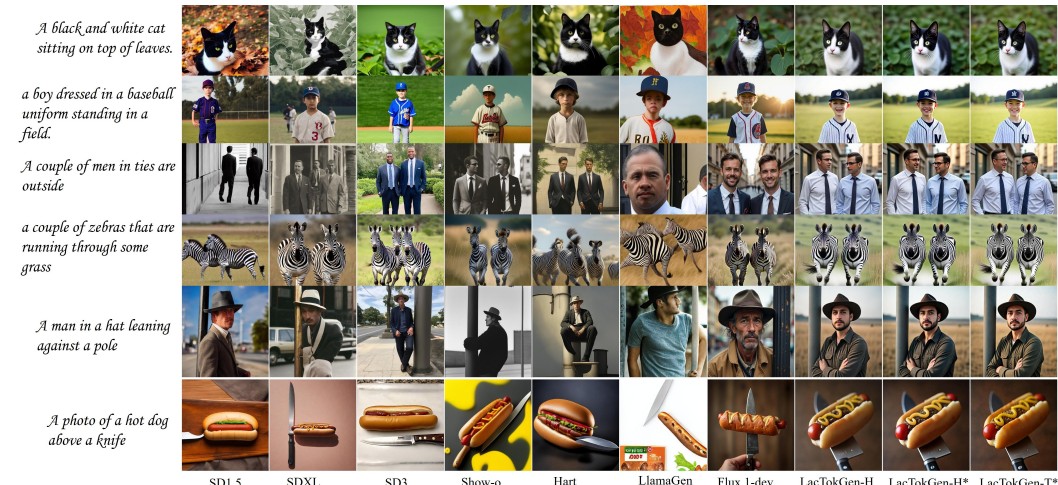

Figure 3: Visual comparison of the generated images by different models.

## 5.3 ABLATION STUDY

As outlined in Table 4, we conduct several experiments on MJHQ-5K to verify the effectiveness of the key components with respect to our LacTok, where LlamaGen is adopted as baseline.

**Latent diffusion decoder.** Compared with LLamaGen, LADD shows worse metrics concerning P, S, L. The reason lies in the fact that LADD predicts the latent representation of the raw image, leading to discrepancies in color and lighting compared to the original image.

**LADD acceleration.** LADD+HyperSD and LADD+TLCM denote that one-step HyperSD and three-step TLCM are directly used to accelerate reconstruction procedure. It can be seen that TLCM outperforms HyperSD. The probable reason is that TLCM is stronger to reconstruct the fine details of the image due to its better generation performance.

**Reconstruction loss.** Through optimizing VQ-LADD using pixel reconstruction loss, LacTok-H surpasses LLamaGen and VQ-LADD by a large margin in terms of rFID, S, and L metrics. The result indicates that the proposed reconstruction loss is critical to improve tokenizer's performance, which is able to enforce the reconstructed image close to raw image in pixel space.

**Token number.** As summarized in Table 5, we can see even using 192 tokens, LacTok-H can reconstruct 1024-pixel image with high performance. As the increase of the token number, the performance is further improved.

**CFG scale.** Table 6 lists the performance of LacTok-T* for text-to-image generation with different CFG scale. It can be observed that LacTok-T* is capable of generating high-quality image using different CFG. The performance can be improved with higher CFG, and the improvement becomes slight when scale>2.

## 6 CONCLUSION

We present LacTok, a discrete tokenizer, which leverages pretrainded LDMs assisted by acceleration models for 1024-pixel image reconstruction with only 256 tokens. LacTok is trained by diffusion loss and pixel reconstruction loss sequentially. It is also extended to text-to-image generation models, LacTokGen, through an autoregressive model. Extensive experiments demonstrate our LacTok outperforms the existing methods for high-resolution image reconstruction. Our LacTokGen shows strong capability to generate high-quality images, achieving 0.73 score on GenEval benchmark.

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

## A    IMPLEMENTATION DETAILS

We adopt the encoder in LLamaGen as our encoder and SDXL as the pretrained LDM. For tokenizer training, diffusion loss is first used to train LacTok for 80000 iterations, and then reconstruction loss is exploited to further optimize LacTok for 20000 iterations. 4 A100 is used with Adam Optimizer to train LacTok, where $\beta_1 = 0.9$, $\beta_2 = 0.99$, learning rate (lr) = 1e-5, total batch size = 4. All the images $\mathbf{x}_0$ with progressive resolution from 512 to 1024 pixel size are encoded into latent space by SDXL-VAE. The images are also randomly down-sampled to $\{224, 256, 288\}$ pixel size, which is then feed into encoder and quantilizer, yielding condition of the LDM. During the training stage of LacTokGen, flan-t5-xl is utilized to extract text features, AR model is initialed by pretrained GPT-XL with 77M parameters in LlamaGen. In this stage, we set lr=1e-4, batch size=320 with 8 A100. AdamW optimizer with $\beta_1 = 0.9$, $\beta_2 = 0.99$ is used to train LacTokGen for 20K iterations. All the tokenizers are trained on ImageNet, except for those specifically marked with an asterisk (*), which are trained on the constructed data.

## B  USAGE OF LLM

We employ DeepSeek to refine our writing.

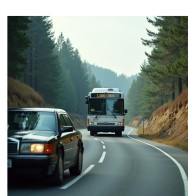
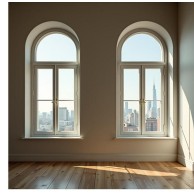
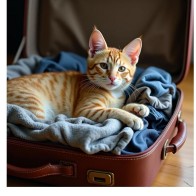

*Caption: A white bus follows a black car on a winding road flanked by tall pine trees in a serene forest setting.*
*Caption: Two arched windows with white frames sit side by side in a beige-walled room, offering a view of a city skyline.*
*Caption: An orange tabby cat rests comfortably on blue and gray clothes inside an open brown suitcase.*

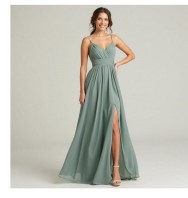
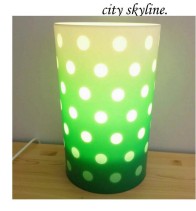
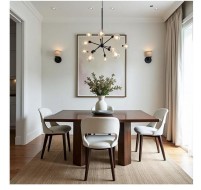

*Caption: A woman wears a long, sage green chiffon bridesmaid dress with a high slit and thin straps, posing gracefully.*
*Caption: A single lamp with an ombre green base and white polka dots sits on a wooden table, emitting soft light.*
*Caption: A chic neutral dining room features a dark wood table, three white chairs, a modern chandelier, and a large abstract painting on the wall.*

Figure 4: The examples of images and their captions used to train LacTokGen.

Table 5: The effect of token number for LacTok-H to reconstruct 1024-pixel image on MJHQ-5K dataset.

| #Tokens | rFID↓ | P↑ | S↑ | L↓ |
|---|---|---|---|---|
| 192 | 12.35 | 18.46 | 0.66 | 0.40 |
| 256 | 11.34 | 19.16 | 0.68 | 0.38 |
| 324 | **11.01** | **19.33** | **0.69** | **0.37** |

Table 6: The effect of CFG scale for LacTokGen-T* to generate image on MSCOCO-2017 5K validation set.

| scale | IR↑ | MPS↑ | HPSv2↑ |
|---|---|---|---|
| 1.5 | 0.86 | 12.37 | 0.303 |
| 2 | 0.89 | 12.38 | **0.304** |
| 3 | **0.90** | 12.40 | **0.304** |
| 7 | **0.90** | **12.41** | **0.304** |

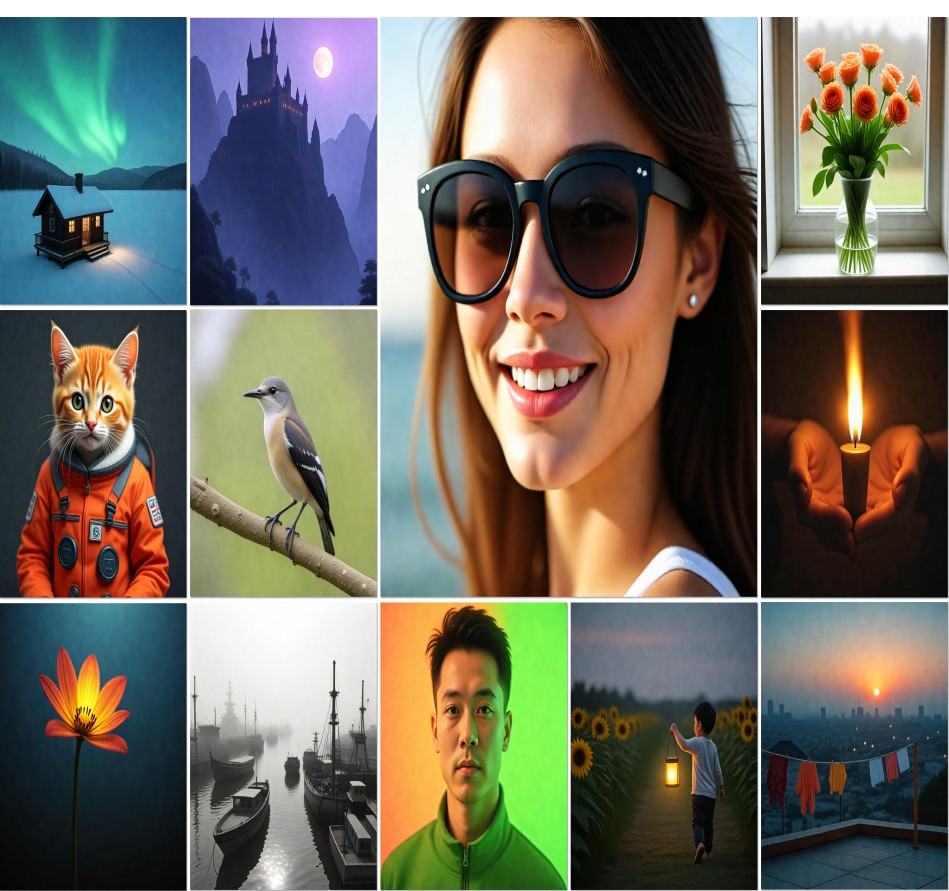

Figure 5: 1024-pixel image generation results of LacTokGen-T* in an autoregressive way with 256 tokens.

