# OpenReview forum: "LacTok: Latent Consistency Tokenizer for High-resolution Image Reconstruction and Generation by 256 Tokens"
_ICLR.cc/2026/Conference — ICLR 2026 Conference Withdrawn Submission_

### Official Review · Reviewer_ATwW · 2025-10-18

**Soundness:** 2
**Presentation:** 2
**Contribution:** 2
**Rating:** 2
**Confidence:** 4

**Summary:**

The paper introduces LacTok, a discrete image tokenizer explicitly aligned to the latent space of a pretrained LDM. The decoder (LADD) is formed by copying LDM blocks and linking them with zero-convolution bridges (ControlNet-style), and then accelerated with HyperSD (1-step) and TLCM (≥2-step) so that pixel-space supervision can flow through a few-step latent path. With 256 tokens for a
1024×1024 image (≈16× fewer tokens than typical VQGAN setups), the method reports strong reconstruction and builds an autoregressive T2I model (LacTokGen) atop these tokens. On COCO-2017 5k, reconstruction reaches rFID = 10.8; on GenEval, LacTokGen is reported around 0.73 with comparisons to both diffusion and AR baselines.

**Strengths:**

1. The “frozen LDM + trainable copy + zero-conv” LADD, combined with few-step consistency, is a clear engineering recipe to make pixel supervision effective.

2. Representing 1024x1024 images with 256 tokens directly eases AR sequence length; the COCO rFID = 10.80 number is competitive and easy to interpret.

3. Compute and training settings are documented (tokenizer on 4×A100, AR on 8×A100), and the paper explicitly discloses LLM assistance in writing.

**Weaknesses:**

1. The abstract claims LacTokGen "surpasses current SOTA" on GenEval, yet FLUX.1-dev in Table 3 is not current SOTA. Should scope the claim (e.g., “among compared models”).

2. The main reconstruction tables (Table 1, Table 2) mix methods with different token budgets and training regimes, while LacTok uses 256 tokens and sometimes constructed data (asterisked). Stronger evidence would include same-token (256) baselines for TiTok/FlexTok/LlamaGen on the same datasets.

3. One–two-step decoders often reduce stochastic detail (colors/textures). The paper lacks quantitative checks such as codebook-usage/entropy and duplication rates, and a structured failure-mode analysis. Please add these to clarify any quality–diversity trade-offs.

4. LacTokGen is trained largely on constructed/synthetic pairs, with IR > 0.9 and MPS > 12 filters to form ~20M pairs. This likely helps GenEval, but it also raises overlap/coverage concerns; a stratified analysis or leakage check would strengthen the claim.

5. The AR model is described as “GPT-XL with 77M parameters in LlamaGen”; the naming (GPT2-XL?)/size (?) pairing is confusing. Please report layers/hidden sizes/vocab and align terminology with LlamaGen.

6. There are no 2K/4K reconstructions at the same 256-token budget and no step–quality–speed curves beyond the HyperSD/TLCM points, which would help justify the few-step design.

**Questions:**

1. Please reconcile the GenEval SOTA statement with Table 3 (or scope it precisely to Table 3); releasing the exact evaluation script and seeds would help.

2. Can you add same-token (256) runs for TiTok/FlexTok/LlamaGen on the same datasets, so token count isn’t a confounder?

3. Could you report codebook usage/entropy, duplication rates, and a short taxonomy of failure cases (e.g., color drift, texture artifacts) for the few-step decoders?

4. Please clarify the actual details of the AR model behind “GPT-XL 77M” (layers, dims, vocab, context) and align naming with LlamaGen.

5. If available, add 2K/4K reconstruction/generation and 1/2/4-step quality–speed curves.

---

### Official Review · Reviewer_HPkr · 2025-10-28

**Soundness:** 2
**Presentation:** 3
**Contribution:** 2
**Rating:** 2
**Confidence:** 3

**Summary:**

The paper presents LacTok (Latent Consistency Tokenzier). The goal is to represent a high-resolution image in as few tokens as possible. It uses 256 discrete tokens to represent an 1024x1024 images, which is 16x more compression than a typical VQGAN. It has a transformer encoder (LlamaGen), and a LADD decoder (diffusion).

It achieves reasonable results in reported benchmarks.

**Strengths:**

(+) The paper is relatively easy to follow. I can follow the author's ideas well

(+) The paper is relatively well motivated. Aligning discrete tokenization with LDM latent space seems make sense

(+) It should theoretically maybe make discrete image generation faster than previous methods given a smaller amount of tokens needed to generate a high-resolution image

**Weaknesses:**

(-) missing computational analysis. The paper claims efficiency but doesn't provide concrete comparisons. Since the decoder is a consistency model, does the method save time end-to-end even though fewer tokens need to be generated?

(-) The paper just tested on using a pretrained LlamaGen encoder and a SDXL decoder. The generalization capability and performance degradation when using different pretrained models is not explored. Not sure if the method proposed in the paper generalizes well or not

(-) Why seedtok and titok is using 64 and 128 tokens but proposed method uses 256 tokens. Titok is one of the closest competitors to this paper, however the experiments in this paper gives a significant disadvantage to it by having only 128 tokens?

(-) The results are not very strong. For example, in Table 1, rFID is much higher than some competitors (e.g. TiTok, even though TiTok is compressed to 128 tokens)

**Questions:**

See weakness section above

---

### Official Review · Reviewer_Fg1K · 2025-10-30

**Soundness:** 3
**Presentation:** 3
**Contribution:** 2
**Rating:** 4
**Confidence:** 3

**Summary:**

This paper proposes LacTok, a discrete tokenizer for efficient compression of high-resolution images, achieving a 16x compression ratio compared with conventional VQGAN.

**Strengths:**

1. The method achieves effective compression and generation of high-resolution images, introducing the Consistency Model to ensure accurate pixel-level reconstruction.

2. Extensive experiments demonstrate the effectiveness of the proposed approach.

**Weaknesses:**

1. This work is largely engineering-oriented. The proposed LacTok combines several existing components, including VQGAN and Consistency Models, but lacks new methodological innovations or conceptual insights.

2. There are potential fairness issues in experimental comparison. The authors synthesize 30 M images and regenerate captions with Qwen2.5-VL-72B for training, while the baselines do not appear to use the same data. It is therefore difficult to disentangle how much of the performance improvement comes from the model architecture versus the training data (see Sec. 4.4).

3. In the GenEval benchmark, the paper only reports results on Two-Obj, Position, and Color-Attri. It would be helpful to also include other dimensions such as Single-Obj, Shape, Spatial, and Non-Spatial for a more comprehensive evaluation.

4. The authors should further clarify why “recent advancements that explore token compression on 256×256 images are difficult to extend to high-resolution cases” (lines 45-48). What are the specific limitations preventing such extension?

**Questions:**

See Weaknesses.

---

### Official Review · Reviewer_wm7T · 2025-10-31

**Soundness:** 3
**Presentation:** 3
**Contribution:** 3
**Rating:** 6
**Confidence:** 3

**Summary:**

This paper introduces the Latent Consistency Tokenizer (LacTok), a novel method for high-resolution image tokenization that aims to solve the trade-off between efficiency and fidelity. The key idea is to bridge discrete visual tokens with the compact latent space of pretrained Latent Diffusion Models (LDMs). To overcome color and brightness discrepancies from a standard LDM decoder, the authors convert it into a "latent consistency decoder," which reduces the multi-step sampling process to 1-2 steps and enables direct pixel-level supervision. This approach allows LacTok to represent 1024x1024 images using only 256 tokens, a 16-fold compression over VQGAN. The authors also present LacTokGen, a text-to-image autoregressive model built on LacTok, which achieves state-of-the-art results on the GenEval benchmark.

**Strengths:**

1. The model achieves a 16-fold compression ratio over standard VQGAN, successfully representing high-resolution 1024x1024 images with only 256 discrete tokens.

2. LacTok excels at reconstructing high-frequency details, such as human faces. This is achieved by its latent consistency decoder, which allows for direct pixel-level supervision.

3. The text-to-image model built upon this tokenizer, LacTokGen, achieves 0.73 on the GenEval benchmark, surpassing both diffusion and other autoregressive models.

**Weaknesses:**

1. Lack of Computational Efficiency. Analysis While the paper's primary claim of achieving a 16x token compression is impressive from a representational standpoint, it lacks a corresponding analysis of computational efficiency. The authors do not provide crucial metrics such as FLOPs, parameters, or inference latency for the tokenizer's encoding and decoding stages. This omission makes it difficult to assess the practical trade-offs of the proposed method.

2. Ambiguity in Baseline Training and Fairness of Comparison The paper's experimental comparison in Tables 1 and 2 raises questions about the training protocol for the baseline models. The caption indicates that most tokenizers were trained by the authors on ImageNet, but the submission lacks sufficient detail about this retraining process. Re-implementing and retraining complex models from other works is a non-trivial task; without using the original authors' official codebases and undergoing a rigorous hyperparameter search, it is possible that the reported baseline performances are suboptimal. This could inadvertently inflate the perceived superiority of LacTok. A more transparent and arguably fairer comparison would involve citing the performance numbers directly from the original papers, provided the evaluation settings are identical. If retraining is necessary due to differing experimental setups, the authors should provide a detailed account of their training methodology for each baseline to assure the community that the comparisons are equitable and reproducible.

**Questions:**

For ICLR paper format, to cite a paper, \citep should be used, instead of \cite.

---

### Note · Authors · 2025-11-12

I have read and agree with the venue's withdrawal policy on behalf of myself and my co-authors.